# Prognostic Impact of Global Longitudinal Strain and NT-proBNP on Early Development of Cardiotoxicity in Breast Cancer Patients Treated with Anthracycline-Based Chemotherapy

**DOI:** 10.3390/medicina59050953

**Published:** 2023-05-15

**Authors:** Gintare Muckiene, Domas Vaitiekus, Diana Zaliaduonyte, Vytautas Zabiela, Raimonda Verseckaite-Costa, Dovile Vaiciuliene, Elona Juozaityte, Renaldas Jurkevicius

**Affiliations:** 1Cardiology Clinic, Medical Academy, Lithuanian University of Health Sciences, 44307 Kaunas, Lithuania; 2Department of Cardiology, Hospital of Lithuanian University of Health Sciences, 50161 Kaunas, Lithuania; 3Kaunas Region Society of Cardiology, 44307 Kaunas, Lithuania; 4Department of Oncology and Hematology, Hospital of Lithuanian University of Health Sciences, 50161 Kaunas, Lithuania; 5Institute of Cardiology, Lithuanian University of Health Sciences, 50161 Kaunas, Lithuania; 6Institute of Oncology, Lithuanian University of Health Sciences, 50161 Kaunas, Lithuania

**Keywords:** anthracycline, doxorubicin, breast cancer, cardiotoxicity, global longitudinal strain, NT-proBNP

## Abstract

*Background*. The most important anthracycline side effect is cardiotoxicity, resulting in congestive heart failure (HF). Early detection of cardiac dysfunction and appropriate treatment can improve outcomes and reduce the progression of HF. The aim of our study was to evaluate changes in clinical data, echocardiographic parameters, and NT-proBNP, as well as their associations with early anthracycline-induced cardiotoxicity (AIC) in patients treated with anthracycline-based chemotherapy. *Methods and Materials*. Patients with breast cancer were prospectively assessed with echocardiography, as well as NT-proBNP testing at baseline, (T0), after two cycles (T1) and four cycles (T2) of chemotherapy. AIC was defined as a new decrease in the LVEF of 10 percentage points, to a value below the lower limit of normal. *Results.* We evaluated 85 patients aged 54.5 ± 9.3 years. After a cumulative dose of 237.9 mg/m^2^ of doxorubicin, 22 patients (25.9%) met the criteria of AIC after chemotherapy. Patients who subsequently progressed to cardiotoxicity had demonstrated a significantly larger impairment in LV systolic function compared to those who did not develop cardiotoxicity (LVEF: 54.0 ± 1.6% vs. 57.1 ± 1.4% at T1, *p* < 0.001, and 49.9 ± 2.1% vs. 55.8 ± 1.6% at T2, *p* < 0.001; GLS: −17.8 ± 0.4% vs. −19.3 ± 0.9% at T1, *p* < 0.001, and −16.5 ± 11.1% vs. −18.5 ± 0.9% at T2, *p* < 0.001, respectively). The levels of NT-proBNP increased significantly from 94.8 ± 43.8 ng/L to 154.1 ± 75.6 ng/L, *p* < 0.001. A relative decrease in GLS ≤ −18.0% (sensitivity: 72.73%; specificity: 92.06%; AUC, 0.94; *p* < 0.001) and a relative increase in NT-proBNP > 125 ng/L (sensitivity: 90.0%; specificity: 56.9%; AUC, 0.78; *p* < 0.001) from baseline to T1 predicted subsequent LV cardiotoxicity at T2. *Conclusions*. Decrease in GLS and elevation in NT-proBNP were significantly associated with AIC, and these could potentially be used to predict subsequent declines in LVEF with anthracycline-based chemotherapy.

## 1. Introduction

Breast cancer is one of the most commonly diagnosed cancers in women, and the incidence of cancer is increasing every year, with more than 1.5 million women worldwide developing breast cancer every year [1]. Since the 1990s, cancer-related mortality has steadily declined [2]. Anthracyclines have been in use since the 1960s. They are effective and widely used in breast cancer treatment, but they have many side effects [3]. The most important anthracycline adverse effect is cardiotoxicity, resulting in congestive heart failure (HF) [4]. Anthracycline-induced HF was first described in 1968 [5]. It is known that anthracycline-induced cardiotoxicity (AIC) may present with symptomatic or asymptomatic cardiotoxicity [6]. Early detection of cardiac dysfunction and appropriate treatment of these patients can improve outcomes and reduce the progression of HF.

Echocardiography is one of the most effective tests to assess cardiac function in cancer patients [6]. It is recommended to evaluate baseline left ventricular ejection fraction (LVEF) in all patients before cardiotoxic cancer treatment initiation and at completion of therapy [6]. A decrease in the LVEF of 10 percentage points, to a value below the lower limit of normal, is defined as cancer therapeutic–related cardiac dysfunction [6,7]. Furthermore, a relative percentage decrease in global longitudinal strain (GLS) of 15% from baseline is considered abnormal and a marker of early LV subclinical dysfunction. Consensus strongly supports a GLS-based follow-up of patients during and after cancer therapy [6,7].

The literature on the use of biomarkers (B-type natriuretic peptide (BNP)/N-terminal pro-brain natriuretic peptide (NT-proBNP)) for AIC risk stratification during cancer therapy still remains controversial. NT-proBNP is one of biomarkers for cardiovascular disease (CVD) risk stratification [8,9].

The purpose of our study was to evaluate the changes in clinical data, echocardiographic parameters, and NT-proBNP, as well as their associations with early cancer therapy-related LV dysfunction in breast cancer patients treated with anthracycline-based chemotherapy.

## 2. Methods and Materials

### 2.1. Study Population

The prospective study was performed during the period 2016–2018 at the Cardiology Department and Oncology and Hematology Department at the Hospital of the Lithuanian University of Health Sciences Kauno Klinikos. A total of 105 patients with breast cancer, who were treated with doxorubicin-based chemotherapy, were screened. Twenty patients were not included due to the exclusion criteria or later declined to participate. The final analytic cohort included 85 patients. A flowchart of patient selection is shown in Figure 1.

Criteria for inclusion for the study:Age > 18 yearsIst–IIIrd stage of breast cancer (non-metastatic disease)Qualification for chemotherapy regimens with conventional doxorubicinNormal systolic LV function (baseline LV EF ≥ 55%), no typical signs of heart failure before the onset of anticancer treatment

Exclusion criteria:Previous radiation therapy, involving the heart and previous chemotherapyKnown significant LV and right ventricular (RV) dysfunction, severe valvular heart disease, arrhythmias, mental illnessContraindication for doxorubicin-based chemotherapyPoor-quality echocardiography windows

A detailed cardiac history (CVD risk factors, concomitant CV medications, family history of premature atherosclerotic CVD (males, age < 55 y; females, age < 65 y)) [10] and clinical evaluation (heart failure symptoms and/or signs, NYHA functional class) were performed by an experienced cardiologist, and an oncologic evaluation was carried out by an experienced oncologist.

Chemotherapy regimens that have been used:AC (doxorubicin plus cyclophosphamide)AC-paclitaxel (doxorubicin plus cyclophosphamide followed by paclitaxel)AC-docetaxel (doxorubicin plus cyclophosphamide followed by docetaxel)FAC (5-FU plus doxorubicin plus cyclophosphamide)TAC (docetaxel plus doxorubicin plus cyclophosphamide)FAC-docetaxel (5-FU plus doxorubicin plus cyclophosphamide followed by docetaxel)

In all regimens, doxorubicin was given by 1 h intravenous infusion. In the beginning, trastuzumab was administered with taxanes, and then, it was given alone after doxorubicin standard treatment in an adjuvant setting for a period of 52 weeks. The present study did not show that the use of trastuzumab had statistically significant effect on LVEF and/or GLS decrease.

### 2.2. Echocardiography

All patients underwent two-dimensional (2D) transthoracic echocardiography and two-dimensional speckle-tracking echocardiography (STE) before initiating anthracycline-based treatment (T0), after two cycles (T1), and after four cycles (T2) of anthracycline-based chemotherapy. All the images were obtained with an ultrasonography system (EPIQ 7, Phillips Ultrasound, Inc., Washington, DC, USA) equipped with S_5-1_ (1 to 5 MHz) fully sampled matrix transducer by one investigator.

Two-dimensional echocardiography was performed according to the current recommendations [11]. The LVEF was assessed using Simpson’s modified biplane method. LV long axis function was studied using the M-mode echocardiography and tissue Doppler imaging (TDI). The mitral annulus peak systolic velocities (S’) and diastolic velocities (E’) were measured in the basal segments of all six walls by TDI, and their averages (S’ and E’ mean) were calculated. The velocities of E- and A-waves were obtained from the transmitral pulsed-wave Doppler. Peak E-wave velocity and TDI peak early diastolic velocity ratio (E/E’) were calculated. Using the M-mode approach, tricuspid annular plane systolic excursion (TAPSE) was measured. Peak tricuspid annulus systolic velocity (S’ RV) by TDI was acquired, according to a standard protocol [11].

We performed an analysis of LV longitudinally, according to the recommendations in [12].

We defined AIC as a new decrease in the LVEF of 10 percentage points, to a value below the lower limit of normal (LVEF < 53%) [7].

### 2.3. Biochemical Markers

Study patients’ blood samples (three milliliters of blood) were taken before chemotherapy (T0), after two cycles (T1), and after four cycles (T2) of doxorubicin-based chemotherapy. Samples were sent to the Department of Laboratory Medicine, at the Hospital of Lithuanian University of Health Sciences, Kauno Klinikos. Tests were performed with standard clinical practice methodology. Human NT-proBNP ELISA kits were used for the plasma NT-proBNP concentration measurements.

### 2.4. The Six-Minute Walking Test

The six-minute walking test (6MWT) was performed, according to the American Thoracic Society comprehensive guidelines [13], as an objective assessment of exercise capacity. The 6MWT was performed in a long straight hospital corridor, over a 30-m distance. The 6-min walk test was selected, and the distance in meters was recorded. Patients were provided with standardized instructions and encouragement. We measured *S*_pO2_, HR responses, symptoms of dyspnea, objective fatigue, and walking distance.

### 2.5. Statistical Analysis

All analyses were performed using Statistical Package for the Social Sciences (SPSS), version 27.0 (Chicago, IL, USA). The categorical variables were shown as absolute numbers and percentages. The continuous variables were presented as means ± standard deviations. For continuous variables with normal distribution, Student’s *t* test was used, and if data were not normally distributed, the Mann-Whitney rank-sum test and Wilcoxon test were used. The Chi-square (χ^2^) or Fisher’s exact test were used to compare proportions. Spearman’s correlation coefficients were used if data were not normally distributed.

The ROC (Receiver Operating Characteristic) curve analysis method was used for determination of optimal values of NT-proBNP, 6MWT.

Logistic regression analyses were used to determine the association between cardiotoxicity and arterial hypertension, family history of CVS, NT-proBNP, and 6MWT via odds ratio (OR) with 95% confidence interval (CI). To estimate the adjusted odds ratio, a multi-variate logistic regression analysis was performed. For all tests, the statistical significance level adopted was *p* ≤ 0.05.

## 3. Results

A total of 85 women with pathologically confirmed breast cancer were enrolled in this study. The study population age ranged from 33 to 75 years old (mean age: 54.5 ± 9.3 years). Patients’ baseline clinical characteristics are presented in Table 1. None of the patients had symptoms or signs of heart failure during the study period. The most common CVD risk factors were AH (42.4%), family history of premature atherosclerotic CVD (27.1%), and dyslipidemia (27.1%). β-blockers and angiotensin-converting enzyme inhibitors/angiotensin receptor blockers were most commonly used in the study population. All the patients had a good exercise capacity before the treatment, and the average 6MWT was 569.6 ± 59.4 m (range 460.0–730.0). The baseline average value of NT-proBNP was normal 94.8 ± 43.8 ng/L (range 55.0–129.0) in the study population.

Distribution of chemotherapy regimens is shown in Table 2. The majority of subjects (63.5% noncardioxicity group and 68.3% cardiotoxicity group) were treated with AC-paclitaxel. All regimens included similar cumulative doses of doxorubicin. Given a comparison of different regiments and medication contributing to the risks for AIC, we did not observe any differences between them. The median doxorubicin cumulative dose in cardiotoxicity group was 239.35 mg/m^2^ (range 150.00–291.00), and noncardiotoxicity was 236.92 mg/m^2^ (range 129.00–303.20), but this difference was not significant.

The baseline 2DE and Doppler parameters (T0, pre-chemotherapy) are summarized in Table 3. All study subjects had normal LVEF (60.6 ± 1.8%, range 56–65) and GLS (−21.1 ± 0.5%, range 20.1–22.6) before treatment. Baseline M-mode and Doppler echocardiography analysis showed normal longitudinal LV systolic function, and LV diastolic function was also normal (Table 3).

All patients underwent clinical evaluation, as well as 2DE and strain analysis by STE after two cycles (T1) and four cycles of doxorubicin-based chemotherapy (T2). A total of 22 patients (25.9%) met the criteria of LV cardiotoxicity after anthracycline-based chemotherapy. A decrease in LVEF, indicative of cardiotoxicity (from 61.8 ± 1.8% at baseline to 49.9 ± 2.1% at T2, *p* < 0.001), was more common three months after the end of the anthracycline therapy (*n* = 18, 81.8%) than that of early cardiotoxicity (*n* = 4, 18.2%). Table 1 shows clinical characteristics, and Table 3 summarizes echocardiographic parameters and their comparisons at baseline after two cycles and four cycles of doxorubicin-based chemotherapy in women who developed cardiotoxicity and those of women who remained without cardiotoxicity (*n* = 63, 74.1%).

There was no significant difference in baseline demographic and clinical characteristics between these two groups, aside from the fact that patients who developed cardiotoxicity had a higher prevalence of AH (68.2%), family history of premature atherosclerotic CVD (50%), and greater use of β-blockers (54.5%) (Table 1).

The baseline LVEF and GLS were within normal limits in both groups. LV end diastolic diameter before chemotherapy was normal in both groups, but it was significantly larger in women who regressed to cardiotoxicity. M-mode and Doppler echocardiographic parameters did not differ between the two groups at baseline (Table 3).

Patients who subsequently progressed to cardiotoxicity during monitoring had demonstrated a significantly larger impairment in LV systolic function (Table 3) compared to those who did not develop cardiotoxicity (LVEF: 54.0 ± 1.6% vs. 57.1 ± 1.4% at T1, *p* < 0.001, and 49.9 ± 2.1% vs. 55.8 ± 1.6% at T2, *p* < 0.001; GLS: −17.8 ± 0.4% vs. −19.3 ± 0.9% at T1, *p* < 0.001, and −16.5 ± 11.1% vs. −18.5 ± 0.9% at T2, *p* < 0.001, respectively). In patients with cardiotoxicity, the mean GLS decreased by 15.2% after two cycles of chemotherapy (from −21.0 ± 0.5% at baseline to −17.8 ± 0.4% at T1, *p* < 0.001), indicating the development of subclinical LV dysfunction in the early period of treatment [7], despite preserved LVEF (54.0 ± 1.6%). In addition, a statistically significant change in GLS was more pronounced (21.4% decrease) in patients with subsequent cardiotoxicity after doxorubicin-based chemotherapy (T2) compared to baseline (*p* < 0.001). Patients who did not develop cardiotoxicity had a smaller LV size at monitoring compared to the cardiotoxicity group (Table 3), and, although LV GLS significantly decreased at T1 follow-up and progressed at T2 follow-up compared to baseline, the reduction was less than 15% (8.5 and 12.3%, respectively).

Conventional parameters of longitudinal LV systolic function, such as mitral annulus plane systolic excursion (MAPSE) and mean peak mitral annulus systolic velocity by TDI (S’), decreased significantly from baseline in both groups (*p* < 0.001) during the study periods, but they were not significantly different between groups, except that MAPSE was lower at T1 follow-up (*p* < 0.05) in patients with cardiotoxicity than for those without cardiotoxicity (Table 3).

The analysis of LV diastolic function showed that LV diastolic function mildly worsened in both groups during the follow-up periods, but it did not differ significantly between the groups after two cycles (T1) and four cycles (T2) of anthracycline-based chemotherapy (Table 3).

The levels of NT-proBNP were measured before the treatment with anthracyclines, after two and four cycles of chemotherapy in the present study, and increased significantly during the monitoring period from 94.8 ± 43.8 ng/L (range 55.0–129.0) at baseline to 154.1 ± 75.6 ng/L (100.0–193.0) at the completion of the anthracycline’s treatment, *p* < 0.001. Women who progressed to cardiotoxicity had significantly higher baseline serum levels of NT-proBNP compared to patients without cardiotoxicity (113.7 ± 37.2 ng/L vs. 87.3 ± 44.3 ng/L, *p* = 0.021) (Table 1). In addition, the increase in NT-proBNP concentrations was more pronounced during the follow-up periods in women who developed cardiotoxicity than in women without cardiotoxicity (171.6 ± 55.3 ng/L vs. 114.1 ± 45.8, *p* < 0.001, at T1, and 207.8 ± 93.2 vs. 133.7 ± 56.5 ng/L, *p* = 0.002, at T2, respectively) (Figure 2). There was a significant moderately positive correlation between serum level of NT-proBNP measured at T1 and cardiotoxicity (r = 0.45, *p* < 0.001).

A statistically significant change in the 6MWT was detected at T1 (*p* < 0.05) and progressed at T2 (*p* < 0.05) in both groups (Figure 3) during monitoring periods. Patients who progressed to cardiotoxicity had slightly reduced exercise tolerance in the 6MWT at baseline compared to patients without cardiotoxicity (541.8 ± 59.0 m vs. 579.3 ± 56.9 m, *p* = 0.001, resp.), and this remained significantly lower after two cycles of chemotherapy (523.9 ± 56.0 m vs. 568.0 ± 58.0 m, *p* = 0.003, resp.) and after four cycles of chemotherapy (508.6 ± 56.1 m vs. 557.0 ± 62 m, *p* = 0.002, resp.). However, in both groups, exercise tolerance, according to the six-minute walking test, remained sufficient and did not reach the limit of impairment. A mild-to-moderate inverse correlation was identified between cardiotoxicity and 6MWT (r = −0.34, *p* = 0.006).

In the univariate analysis, GLS, NT-proBNP, 6MWT, AH, and a family history of CVD were associated with subsequent LV cardiotoxicity *p* < 0.001 (Table 4), and, therefore, they were included in the multivariate logistic regression model. GLS and NT-proBNP were identified as independent predictors of an early development of cardiotoxicity in breast cancer patients treated with anthracycline-based chemotherapy.

The area under the curve (AUC) of the ROC curves for GLS and NT-proBNP as independent predictors for an adverse outcome was 0.94, with *p* < 0.001 (95% CI: 0.868–0.980) and 0.78, as well as *p* < 0.001 (95% CI: 1.852–21.096), respectively. A relative decrease in GLS ≤ −18.0% (sensitivity: 72.73%; specificity: 92.06%) and a relative increase in NT-proBNP > 125 ng/L (sensitivity: 90.0%; specificity: 56.9%) were able to discriminate patients with or without LV cardiotoxicity (Figure 4A,B).

## 4. Discussion

This study showed an early incidence of cancer therapy-related cardiotoxicity in breast cancer women treated with anthracycline-based chemotherapy before therapy with trastuzumab. Detection of early cardiotoxicity in this study was based on clinical assessment and echocardiography during the treatment with doxorubicin-based chemotherapy. In the present study, 22 of the 85 patients who received anthracycline-based chemotherapy exhibited an early LV dysfunction with no HF symptoms or signs during the monitoring period. The main findings of our study were: (1) asymptomatic LV systolic dysfunction occurred in many patients treated with anthracycline-based chemotherapy, particularly in patients with more common CV risk factors, and LV GLS was the parameter that was the earliest and most markedly impaired during chemotherapy; (2) LV GLS measured during doxorubicin-based chemotherapy was associated with the subsequent development of early cardiotoxicity based on LVEF criteria and may identify patients who will develop cardiotoxicity; (3) during anthracycline-based chemotherapy, the increase in serum NT-proBNP levels were more pronounced in patients with cardiotoxicity than without cardiotoxicity, and these were effective diagnostic indicators for predicting early LV cardiotoxicity; (4) exercise capacity in the 6MWT decreased in all patients during the study with chemotherapy, but the decrease was more pronounced in patients with than without cardiotoxicity and was associated with the development of LV cardiotoxicity.

Anthracyclines are a class of chemotherapies that include doxorubicin, widely used in the treatment of various solid and liquid tumors [14]. Doxorubicin-induced cardiotoxicity may occur through a variety of mechanisms, such as interaction with iron, activity of intracellular or intramitochondrial oxidizing enzymes, alteration of endothelin-1 expression in cardiomyocytes, and binding to topoisomerases [15,16,17,18,19,20]. AIC can manifest as asymptomatic LV dysfunction in up to 57% of treated patients [21,22], and symptomatic congestive HF can manifest in up to 16–20% of patients [14,23,24]. In clinical practice, however, these percentages may be even higher.

It is known that AIC is a dose-dependent and cumulative process at a total doxorubicin dose of 450 mg/m^2^ [14]. Billingham et al.’s study results of endomyocardial biopsies showed histopathological changes with doses as low as 240 mg/m^2^, suggesting that subclinical cardiotoxicity may be present as early as the first dose [25]. The present study also found that, though the mean cumulative doxorubicin dose was 233.9 ± 29.6 mg/m^2^ (ranging from 129.0 to 303.2) quite low, early subclinical LV systolic dysfunction was still observed in some patients in the study population, as is indicated in the literature [25].

Early subclinical cardiotoxicity diagnosis and treatment are determinant to improve prognosis. Therefore, early diagnosis of asymptomatic LV dysfunction is very important. Transthoracic echocardiography is the preferred imaging technique [26]. AIC diagnosis is based on a reduction in LVEF [6,7]. However, the LVEF measurement is a relatively insensitive tool for detecting cardiotoxicity at an early stage. This is mainly because significant changes in LVEF occur when a large amount of myocardial damage has taken place, and compensatory mechanisms are exhausted [27]. Deformation imaging by STE improves assessment of LV performance, yielding functional and prognostic information distinct from LVEF [28,29]. Kalam et al. published a large systematic meta-analysis (5721 adults) of GLS and LVEF with HF, acute myocardial infarction, and valvular and miscellaneous cardiac diseases, and they found that GLS was a more sensitive prognosticator of outcome compared to LVEF [26]. GLS has been reported to be more accurate with regard to LVEF in the detection of incipient and/or subclinical HF, for example, cancer therapy HF [28]. Consistent with previous studies [29,30,31], the present study showed that all patients treated with anthracycline-based chemotherapy had an early reduction in LV GLS, despite preserved LVEF, and it was more impaired after treatment in patients with cardiotoxicity. In patients with established cardiotoxicity, subclinical LV dysfunction was detected early—after two cycles of chemotherapy based on LV GLS change from baseline. Furthermore, LV GLS measured after two cycles of chemotherapy was able to independently predict subsequent cardiotoxicity. The ROC curve of LV GLS was plotted, and the AUC was 0.94 (95% CI: 0.868–0.980), and it was also concluded that the optimal GLS threshold was −18.3%, the diagnostic sensitivity of cardiotoxicity was 95.5%, and the specificity was 87.3%.

In recent years, natriuretic peptides have been widely used for the diagnosis of early doxorubicin-induced cardiotoxicity. BNP is a rapid and accurate indicator of HF, and it has a half-life of 15 to 20 min. NT-proBNP has a half-life of 60–120 min [32,33]. Because NT-proBNP has a long half-life and is stable, it can accumulate at higher concentrations, and therefore symptomatic and asymptomatic LV dysfunction can be diagnosed [33].

The combination of cardiac biomarkers and GLS is believed to increase diagnostic accuracy in early cardiotoxicity. The literature on the use of biomarkers for cardiotoxicity risk stratification before cancer therapy is still limited. It is recommended to measure cardiac serum biomarkers, such as cardiac troponin (cTn) I or T and BNP/NT-proBNP during cancer therapies, including anthracyclines and others [6,7]. Elevated NPs have the potential to identify early doxorubicin-induced cardiotoxicity because their rise can detect LV dysfunction earlier than echocardiography [34].

Feola et al. followed 53 breast cancer patients treated with anthracycline-based chemotherapy and found that patients had a reduction in LVEF of >10% at follow-up, with a baseline BNP of 55.5 ± 72.3 pg/mL, and those who did not had a baseline of 26.1 ± 21.4 pg/mL (*p* = 0.07 HR 0.96–1) (*p* = 0.07 HR 0.96–1) [9]. Other studies that investigated the association between BNP/NT-proBNP and cardiotoxicity showed different findings. Dodos et al. followed 100 patients who received anthracycline-based chemotherapy, and they did not find correlation between baseline NT-proBNP and LVEF reduction [35]. Sawaya et al. showed similar results—NT-proBNP did not predict cardiotoxicity [29].

The present study evaluated changes in serum NT-proBNP levels during anthracycline-based chemotherapy and the association with cardiotoxicity in breast cancer patients. In our study, patients with progressive cardiotoxicity had higher baseline serum levels of NT-proBNP compared to patients without cardiotoxicity, suggesting that some of the study patients already had increased biological stress and myocardial strain [34] due to preexisting CV risk factors. In addition, increased concentrations of NT-proBNP were observed following anthracycline-based chemotherapy and higher NT-proBNP levels in the cardiotoxicity group compared to the non-cardiotoxic group, suggesting that increase was induced by the anthracycline-based treatment. The serial evaluation of NT-proBNP concentration in patients treated with doxorubicin-based chemotherapy showed a predictive value for the occurrence of cardiotoxicity. The area under the curve (AUC) was 0.78 (95% CI: 1.852–21.096), suggesting that the NT-proBNP level during the treatment with anthracycline had a role in diagnosing cardiotoxicity. ROC curve analysis revealed that the optimal NT-proBNP threshold was 125 ng/L. The diagnostic sensitivity of AIC was 90.0%, and the specificity was 56.9%.

6MWT is a widely available, easily performed, and well tolerated test for assessing the functional capacity of patients with HF. The benefit of 6MWT results in the diagnosis of HF has been studied in patients with HF [36], coronary artery disease [37], and cancer [38]. However, little is currently known about 6MWT and cardiotoxicity. One of our research objectives was to identify the role of 6MWT as an early predictor of cardiotoxicity in patients treated with anthracycline-based chemotherapy. However, although 6MET was associated with cardiotoxicity, it was not an independent predictor of cardiotoxicity in a multivariate analysis of a logistic regression model. This may have been due to the fact that none of the patients developed symptomatic LV dysfunction during the study.

One limitation of our study is that it is a relatively small sample. However, statistically significant correlation demonstrated that LV GLS and NT-proBNP may be used as independent predictors of doxorubicin-induced subclinical LV cardiotoxicity.

Another limitation of our study is that the chemotherapy regimen and the concomitant medication are nonhomogeneous. Most of our patients were treated with AC-P regimen, and all regimens used a similar cumulative dose of doxorubicin. It is known that some drugs used in breast cancer treatment regimens may contribute to cardiotoxicity, and we compared different regimens and did not find any difference in AIC [39]. Cardiotoxicity is considered the most important adverse reaction of doxorubicin-based chemotherapy in patients with breast cancer. Some drugs used in chemotherapy regimens have been shown to be associated with cardiac dysfunction. Therefore, our study results show that anthracycline-based chemotherapy, and not doxorubicin alone, may possibly contribute to AIC in breast cancer patients [40,41].

## 5. Conclusions

Despite their cardiotoxicity, anthracyclines are some of the most important drugs for breast cancer treatment. Therefore, while using these drugs, the risk factors associated with cardiotoxicity should be considered, and cardiac function should be accurately measured. Our study demonstrated that LV GLS and NT-pro BNP may be used as independent predictors of anthracycline-induced subclinical LV cardiotoxicity. Further studies are needed to establish more accurate baseline cut-off values for NT-proBNP and 6MWT in cancer patients, and they would be more suitable for the early diagnosis of cardiotoxicity.

## Figures and Tables

**Figure 1 medicina-59-00953-f001:**
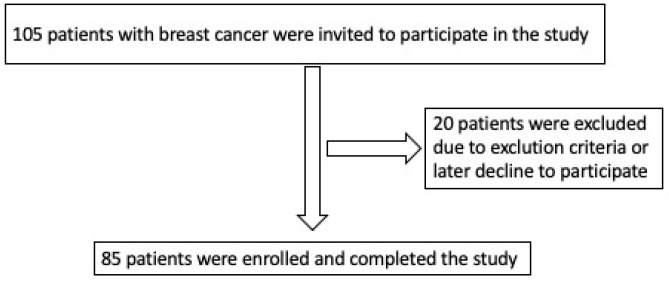
A flowchart of patient selection.

**Figure 2 medicina-59-00953-f002:**
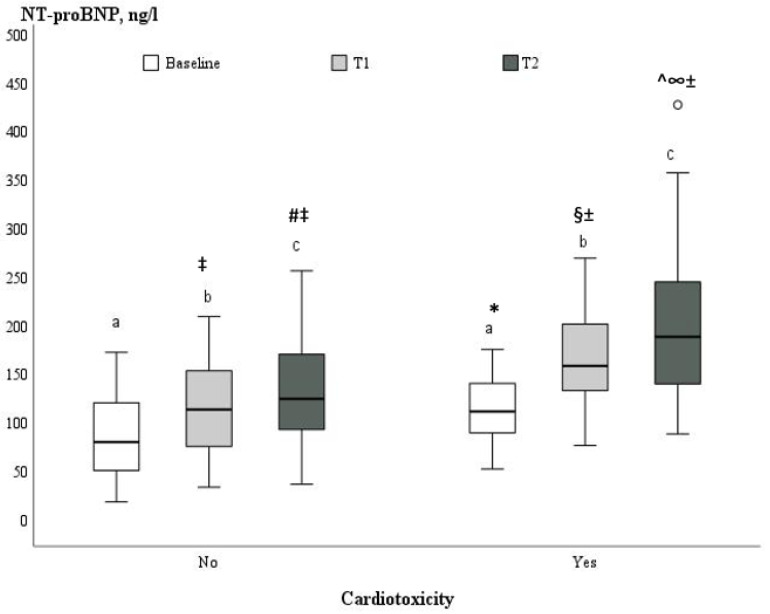
Boxplot illustrating NT-proBNP serum values at baseline (a), T1 (b) and T2 (c) between groups with and without cardiotoxicity. * *p* < 0.05 vs. no cardiotoxicity at baseline (T0); ^§^
*p* < 0.05 vs. no cardiotoxicity at T1; ^±^
*p* < 0.05 vs. cardiotoxicity at baseline; ^ *p* < 0.05 vs. no cardiotoxicity at T2; ^∞^
*p* < 0.05 vs. cardiotoxicity at T1; ^‡^
*p* < 0.05 vs. no cardiotoxicity at baseline; ^#^
*p* < 0.05 vs. no cardiotoxicity at T1.

**Figure 3 medicina-59-00953-f003:**
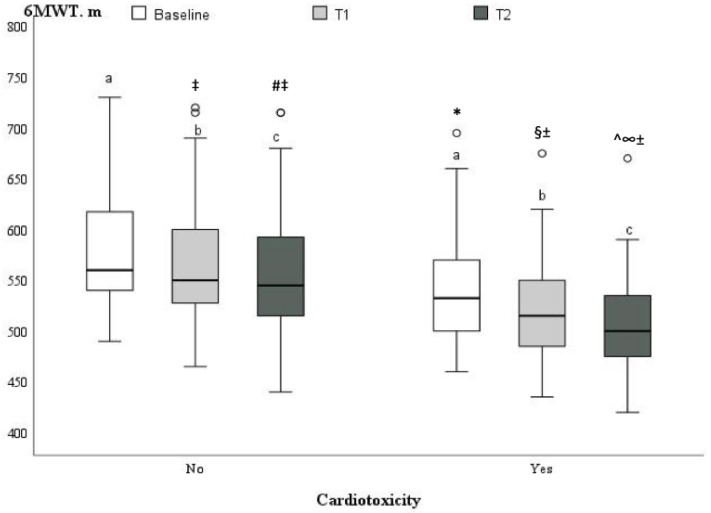
Boxplot, illustrating the six-minute walking test distances at baseline (a), T1 (b) and T2 (c) between the groups with and without cardiotoxicity. ^‡^
*p* < 0.05 vs. no cardiotoxicity at baseline; ^#^
*p* < 0.05 vs. no cardiotoxicity at T1; * *p* < 0.05 vs. no cardiotoxicity at baseline (T0); ^§^
*p* < 0.05 vs. no cardiotoxicity at T1; ^±^
*p* < 0.05 vs. cardiotoxicity at baseline; ^ *p* < 0.05 vs. no cardiotoxicity at T2; ^∞^
*p* < 0.05 vs. cardiotoxicity at T1.

**Figure 4 medicina-59-00953-f004:**
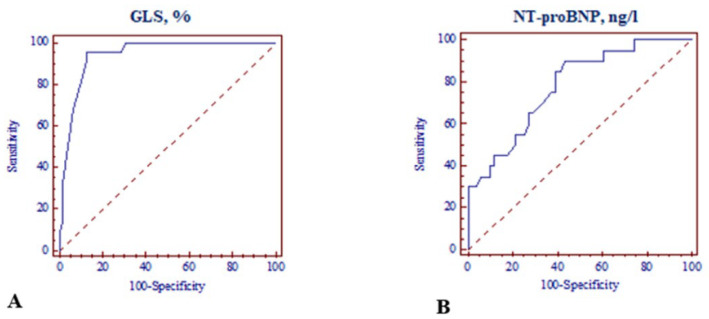
ROC curves for LV global longitudinal strain and NT-proBNP in predicting LV cardiotoxicity. (**A**) ROC curve analysis of LV GLS; (**B**) ROC curve analysis of NT-proBNP. ROC: receiver operating characteristic; LV: left ventricular; GLS: global longitudinal strain; NT-proBNP: N-terminal pro B-type natriuretic peptide.

**Table 1 medicina-59-00953-t001:** Baseline clinical characteristics in patients with or without cardiotoxicity. Values are expressed as means ± SDs or as a number (percentage). The *p* values compare cardiotoxicity to no cardiotoxicity.

	All Patients (*n* = 85)	Noncardiotoxicity (*n* = 63; 74.1%)	Cardiotoxicity (*n* = 22; 25.9%)	*p*-Value
Age, yrs	54.5 ± 9.3	54.4 ± 8.5	54.6 ± 11.4	0.77
BMI, kg/m^2^	27.7 ± 5.5	27.8 ± 5.4	27.5 ± 5.8	0.79
*CVD risk factors*				
AH, *n* (%)	36 (42.4)	21 (33.3)	15 (68.2)	0.004
Diabetes mellitus, *n* (%)	16 (18.8)	10 (15.9)	6 (27.3)	0.24
Smoking, *n* (%)	17 (20.0)	12 (19.1)	5 (22.7)	0.26
Family history of CVD, *n* (%)	23 (27.1)	12 (19.1)	11 (50.0)	0.005
Dyslipidemia, *n* (%)	23 (27.1)	14 (22.2)	9 (40.9)	0.89
*Medications*				
ACE inhibitors/ARBs	24 (28.2)	16 (25.4)	8 (36.4)	0.33
β-blockers	26 (30.6)	14 (22.2)	12 (54.5)	0.005
Diuretics	6 (7.1)	3 (4.8)	3 (13.6)	0.18
Calcium channel blockers	8 (9.4)	4 (6.4)	4 (18.2)	0.10
NT pro-BNP (ng/L)	94.8 ± 43.8	87.3 ± 44.3	113.7 ± 37.3	0.021
6MWT, m	569.6 ± 59.4	579.3 ± 56.9	541.8 ± 59.0	0.005

BMI: body mass index; AH: arterial hypertension; CVD: cardiovascular disease; Family history of CVD: family history of premature atherosclerotic cardiovascular disease; ACE inhibitors: angiotensin-converting enzyme inhibitors; ARB: angiotensin receptor blockers; β-blockers: beta blockers; NT pro-BNP: N-terminal pro B-type natriuretic peptide; 6MWT: 6-min walking test.

**Table 2 medicina-59-00953-t002:** Distribution of chemotherapy regimens. The *p* values compare cardiotoxicity with no cardiotoxicity.

Regimen	All Patients (*n* = 85)	Noncardiotoxicity (*n* = 63; 74.1%)	Cardiotoxicity (*n* = 22; 25.9%)	*p*-Value
AC	8 (9.4)	6 (9.5)	2 (9.1)	1.00
AC-paclitaxel	55 (64.7)	40 (63.5)	15 (68.3)	0.692
AC-docetaxel	8 (9.4)	7 (11.1)	1 (4.5)	0.674
FAC-docetaxel	6 (7.1)	5 (7.9)	1 (4.5)	1.00
TAC	5 (5.9)	2 (3.2)	3 (13.6)	0.107
FAC	3 (3.5)	3 (4.8)	0 (-)	0.565
Doxorubicin cumulative dose (mg/m^2^)	233.9 ± 29.6	234.6 ± 30.1	231.3 ± 28.5	0.80

A—Doxorubicin, C—Cyclophosphamide, F—5-Fluorouracil, T—Docetaxel.

**Table 3 medicina-59-00953-t003:** Two-dimensional echocardiographic parameters before and after chemotherapy in patients with and without cardiotoxicity.

Variables	T0				T1			T2		
	All Patients (*n* = 85)	Cardiotoxicity			Cardiotoxicity			Cardiotoxicity		
		No (*n* = 63; 74.1%)	Yes (*n* = 22; 25.9%)	*p*-Value	No (*n* = 63; 74.1%)	Yes (*n* = 22; 25.9%)	*p*-Value	No (*n* = 63; 74.1%)	Yes (*n* = 22; 25.9%)	*p*-Value
LVEF (%)	60.6 ± 1.8	60.1 ± 1.6	61.8 ±1.8	<0.001	57.1 ± 1.4 ^±^	54.0 ± 1.6 *	<0.001	55.8 ± 1.6 ^±^^	49.9 ± 2.1 ^*§^	<0.001
GLS (%)	−21.1± 0.5	−21.1 ± 0.5	−21.0 ± 0.5	0.52	−19.3 ± 0.9 ^±^	−17.8 ± 0.4 *	<0.001	−18.5 ± 0.9 ^±^^	−16.5 ± 11.1 ^*§^	<0.001
LVEDD (mm)	46.1 ± 3.9	45.9 ± 4.2	46.8 ± 2.6	0.018	46.2 ± 3.8 ^±^	47.1 ± 3.0	0.04	46.6 ± 3.8 ^±^^	47.1 ± 3.9	0.019
LVEDD index (mm/m^2^)	25.1 ± 2.8	24.8 ± 2.7	25.6 ± 2.8	0.17	25.0 ± 2.6 ^±^	25.7 ± 2.9	0.21	25.5 ± 3.7 ^±^^	25.8 ± 3.0	0.20
MAPSE (mm)	14.8 ± 1.9	14.7 ± 2.0	14.9 ± 1.4	0.68	13.9 ± 1.6 ^±^	13.1 ± 1.0 *	0.043	13.3 ± 1.7 ^±^^	13.0 ± 1.8 *^§^	0.30
S’ mean (cm/s)	9.0 ± 1.3	9.1 ± 1.3	8.9 ± 1.2	0.98	8.4 ± 1.1 ^±^	7.9 ± 1.1 *	0.08	8.0 ± 1.2 ^±^^	7.6 ± 1.1 *^§^	0.18
E (cm/s)	72.7 ± 16.7	72.5 ± 15.6	73.2 ± 19.8	0.50	70.0 ±13.8 ^±^	69.4 ±12.4 *	0.85	67.3 ± 14.0 ^±^^	66.3 ± 11.3 *	0.85
A (cm/s)	73.7 ± 17.7	75.0 ± 17.9	70.1 ± 17.2	0.29	76.2 ± 14.7	73.6 ± 14.8	0.46	79.0 ± 17.6 ^±^	74.0 ± 17.7	0.30
E/A ratio	1.07 ± 0.4	1.04 ± 0.4	1.15 ± 0.3	0.08	0.96 ± 0.3 ^±^	0.98 ± 0.3 *	0.40	0.89 ± 0.3 ^±^^	0.95 ± 0.3 ^*§^	0.29
E’ mean (cm/s)	11.4 ± 2.3	11.4 ± 2.4	11.6 ± 2.1	0.88	10.3 ± 2.1 ^±^	10.3 ± 1.9 *	0.86	9.7 ± 2.1 ^±^^	10.0 ± 2.4 ^*§^	0.98
E/E’ ratio	6.7 ± 1.4	6.7 ± 1.4	6.9 ± 1.3	0.79	6.9 ± 1.3 ^±^	6.8 ± 1.0	0.22	7.3 ± 1.6 ^±^^	7.2 ± 1.4	0.89

Values are expressed as means ± SDs. The *p* values compare cardiotoxicity with no cardiotoxicity. * *p* < 0.05 vs. cardiotoxicity at baseline (T0); ^§^
*p* < 0.05 vs. cardiotoxicity at T1; ^±^
*p* < 0.05 vs. no cardiotoxicity at baseline (T0); ^ *p* < 0.05 vs. no cardiotoxicity at T1. LVEF: left ventricle ejection fraction; GLS: global longitudinal strain; LVEDD: left ventricle end-diastolic diameter; MAPSE: mitral annular plane systolic excursion; S’: mitral annular peak systolic velocity, E’: peak early diastolic transmitral flow velocity; A: peak late (atrial) diastolic transmitral flow velocity; E’: peak mitral annular tissue velocity during early filling; E/E: peak early diastolic transmitral flow velocity to peak early mitral annular tissue velocity ratio; T0–before chemotherapy; T1—after two cycles of doxorubicin-based chemotherapy; T2—after four cycles of doxorubicin-based chemotherapy.

**Table 4 medicina-59-00953-t004:** Univariate and multivariate logistic regression analyses for the predictors of cardiotoxicity.

Variable	Univariate		Multivariate	
	OR [95% CI]	*p*-Value	OR [95% CI]	*p*-Value
GLS ≤ −18.0%	30.933 [8.350–114.4590]	<0.001	12.849 [2.783–59.326]	0.001
NT-proBNP > 125 ng/L	11.864 [2.487–56.595]	<0.001	7.091 [1.034–48.642]	0.046
Family history of CVD	4.250 [1.493–12.095]	0.005	2.874 [0.615–13.417]	0.18
AH	4.286 [1.517–12.112]	0.004	1.299 [0.300–18.105]	0.73
6MWT ≤ 515 m	7.200 [2.405–21.554]	<0.001	3.704 [0.758–9.777]	0.11

CI: Confidence interval; OR: Odds ratio; GLS—Global longitudinal strain; NT pro-BNP: N-terminal pro B-type natriuretic peptide; Family history of CVD: Family history of premature atherosclerotic cardiovascular disease (males, age < 55 y; females, age < 65 y); AH: Arterial hypertension; 6MWT: 6-min walking test.

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
