# Peer review of "Prognostic Impact of Global Longitudinal Strain and NT-proBNP on Early Development of Cardiotoxicity in Breast Cancer Patients Treated with Anthracycline-Based Chemotherapy"

_medicina, 2023, doi:10.3390/medicina59050953_

Round 1

Reviewer 1 Report

In the study, authors aimed to evaluate the changes in clinical data, echocardiographic parameters and NT-proBNP, and their associations with early cancer therapy-related LV dysfunction in patients with breast cancer treated with athracyclines-based chemotherapy.  However, as given in chemotherapy regime, various chemotherapeutics were also applied to patients which also has cardiotoxic effects. (These are regiments of chemotherapy that have been used for treatment: AC (Doxorubicin 60 mg/sqm IV plus cyclophosphamide 600 mg/sqm IV on day 1 every 3 week for four cycles), FAC (5-FU 500 mg/sqm IV on days 1 and 8 or days 1 and 4 plus doxorubicin 50 mg/sqm IV on day 1 plus cyclophosphamide 500 mg/sqm IV on day 1 every 3 week for six  cycles),  ………)

How did the authors understand that all these effects were due to athracyclines-based therapy? Otherwise, they should have selected groups which used only other type of chemotherapeutics in order to avoid the false positive results. Moreover, it was not clearly presented that all these different treatment regimes affected the results or not. This section must be written and explained in results section clearly.

Minor editing of English language required. There are only some typos.

Reviewer 2 Report

Overall, the study is well-designed and provides important insights into the early prediction of anthracycline-induced cardiotoxicity in patients with breast cancer. The study's limitations include a relatively small sample size and the lack of a control group. Despite these limitations, the study's findings have the potential to inform clinical practice and improve patient outcomes.

Reviewer 3 Report

Muckiene et al in this manuscript investigated echocardiographic parameters and NT-proBNP, and their associations with early anthracycline-induced cardiotoxicity in patients treated with anthracycline, and found that decrease of GLS and elevation in NT-proBNP were significantly associated with anthracycline-induced cardiotoxicity, and could potentially be used to predict subsequent declines in LVEF with anthracycline-based chemotherapy. Overall, most of data are convincing and supporting major conclusion, there is one concern that should be addressed.

In the table 1, arterial hypertension, family history of cardiovascular disease, beta-blockers, NT pro-BNP and 6-minute walking test are significantly between non-cardiotoxicity and cardiotoxicity groups, what are the differences about these parameters among T0, T1 and T2?

Round 2

Reviewer 1 Report

The manuscript can be accepted in its current form.